# On the Accuracy of uRANS and LES-Based CFD Modeling Approaches for Rotor and Wake Aerodynamics of the (New) MEXICO Wind Turbine Rotor Phase-III

**Shantanu Purohit, Ijaz Fazil Syed Ahmed Kabir**  **and E. Y. K. Ng \*** 

School of Mechanical and Aerospace Engineering, Nanyang Technological University, 50 Nanyang Avenue, Block N3, Singapore 639798, Singapore; shantanu006@e.ntu.edu.sg (S.P.); ijaz0001@e.ntu.edu.sg (I.F.S.A.K.)
* Correspondence: mykng@ntu.edu.sg

**Abstract:** This work presents a comparison study of the CFD modeling with two different turbulence modeling approaches viz. unsteady RANS and LES, on a full-scale model of the (New) MEXICO rotor wind turbine. The main emphasis of the paper is on the rotor and wake aerodynamics. Simulations are carried out for the three wind speeds considered in the MEXICO experiment (10, 15, and 24 ms$^{-1}$). The results of uRANS and LES are compared against the (New) MEXICO experimental measurements of pressure distributions, axial, radial, and azimuth traverse of three velocity components. The near wake characteristics and vorticity are also analyzed. The pressure distribution results show that the LES can predict the onset of flow separation more accurately than uRANS when the turbine operates in the stall condition. The LES can compute the flow structures in wake significantly better than the uRANS for the stall condition of the blade. For the design condition, the mean absolute error in axial and radial velocity components along radial traverse is less than 10% for both the modeling approaches, whereas tangential component error is less than 2% from the LES approach. The results also reveal that wake recovers faster in the uRANS approach, requiring further research of the far wake region using both CFD modeling approaches.

**Keywords:** wind turbine; large eddy simulation (LES); unsteady Reynolds-Averaged Navier-Stokes (uRANS) simulation; rotor aerodynamics; wake aerodynamics; vorticity

## 1. Introduction

Wind energy is becoming an important source of renewable energy. Due to low power density, wind turbines are clustered in wind farms to maximize energy output. In fact, an important limitation of such arrangement is the wake shed by the upstream turbines on the downstream turbines, which leads to lower power production and increased loads on the downstream wind turbines [1,2]. A wind turbine wake has flow scales ranging from millimeters to several hundred meters. Therefore, it necessitates studying the dynamics of wind turbine wakes to design next-gen wind turbines and wind farms.

To design efficient and reliable wind turbines, it is vital to have a proper understanding of rotor and wake aerodynamics. The two most popular experimental studies conducted to examine these effects are NREL Phase VI wind turbine experiment [3] and MEXICO rotor experiments [4,5]. Several authors have used these studies to validate their computational models. Accordingly, a multitude of numerical studies has been done on the MEXICO rotor. Bechmann et al. [6] employed EllipSys3D CFD code, a RANS-based flow solver with *k-ω* SST turbulence model, to validate the MEXICO rotor measurements and extracted 3-D airfoil coefficients from the numerical simulations. The same flow solver has been applied to study the near wake of the MEXICO rotor [7] and validation of the Phase-II MEXICO results [8]. The actuator line method integrated into EllipSys3D is also used to study the near wake of the MEXICO rotor [9,10]. Nathan et al. [11] compared the results of the Phase-II MEXICO rotor with actuator line modeling using OpenFOAM and EllipSys3D

and found that both of these flow solvers predict lower speed wake measurements well but fail when the flow becomes separated. Other researchers have used the MEXICO rotor to explain the rotational effects at the inboard region which leads to stall delay [12,13]. Carrión et al. [14] proposed an all-Mach scheme for flow around wind turbines and used the MEXICO rotor for validation. The wake velocity results obtained from this scheme proved to be consistent with the experimental measurements. Recently, Li et al. [15] used the MEXICO rotor and employed a Lattice Boltzmann/ LES method for the investigation of wake characteristics. The evolution of velocity and wake characteristics agreed well with the experiments.

Due to the computational cost associated with the LES, it is commonly used in indirect rotor modeling with the actuator line and actuator disk methods [16]. Several other CFD studies have been done on the MEXICO rotor [17–24]. The RANS-based flow modeling approaches have been widely studied in the literature. Regodeseves et al. [17] conducted unsteady RANS simulations of the MEXICO rotor including the effects of nacelle and tower and found good agreement of numerical computations against the experiments. More recently, Zhang et al. [18] compared RANS and DDES simulations for the stall condition of the MEXICO rotor and found improved load prediction in the outboard region of the blade compared to those obtained from the RANS. Tsalicoglou et al. [22] studied uniform and yawed inflow conditions and Rodrigues and Lengsfeld [23] investigated the axial conditions of the MEXICO rotor using the RANS approach. Qian et al. [24] compared the RANS and DES numerical computation approaches for the yaw condition of the MEXICO rotor. Johansen et al. [25] used DES to study parked conditions and sinusoidal pitching of the NREL Phase VI wind turbine about its axis. Amiri et al. [19] assessed the performance of different Reynolds Stress Transport (RST) models to predict wind turbine aerodynamics at the stalled condition of the NREL Phase VI wind turbine. The computational results suggest that out of several RST models, the quadratic pressure-strain and elliptic bending RST models perform better for the entire wind speed spectrum considered for the analysis. Hsu et al. [20] used FEM to study the aerodynamics of the NREL Phase VI wind turbine using ALE-VMS formulation and applied the same discrete formulation for all the wind speed cases. The proposed formulation predicted the flow solution accurately, thus, circumventing the need to adjust the RANS-based turbulence model for different flow regimes. Kabir and Ng [26] applied the uRANS method and modeled the rotating effect with a sliding mesh approach to improve the BEM method.

Hybrid RANS and LES techniques bridge the gap between computationally expensive LES and less accurate RANS modeling. It solves the attached boundary layers using the RANS approach and separated flows using the LES approach. This is extensively used in modeling flow through the horizontal axis wind turbines (HAWTs) [27], but little or no improvement over unsteady RANS is observed for the prediction of blade characteristics [19,25,28–30].

As seen from the above literature review, a host of numerical studies on HAWTs are based on the RANS approach. To the best of the authors' knowledge, LES is not used widely in full rotor modeling. Mo et al. [21] used large-eddy simulations to study the wake characteristics of the NREL Phase VI wind turbine. It is noted that no study employing LES has been conducted for the MEXICO rotor that gives a detailed comparison of both rotor and wake aerodynamics. This paper critically assesses the aerodynamic performance of the (New) MEXICO rotor using uRANS and LES. It is a well-known fact that the spatio-temporal wake structure of the turbulence can be better captured by the LES method [31]. This is investigated here for the (New) MEXICO rotor. The pressure distribution is compared at the five radial locations of $\frac{r}{R}$ = 0.25, 0.35, 0.60, 0.82, and 0.92. The results are further complemented with the wake velocity comparison downstream of the blade. Here, axial, radial, and azimuthal traverses of velocity components are also compared against the experimental measurements. A discussion on wake velocity and vorticity contours from the two approaches is also presented.

## 2. Numerical Framework

This section describes the CFD analysis setup of a wind turbine for predicting its aerodynamic performance. This section begins with the description of the MEXICO experiment and the computational geometry used for the analysis, followed by the mesh details and boundary conditions applied to the domain. The governing equations used in uRANS based $k$-$\omega$ SST model and WALE-based LES are highlighted next. Finally, a short discussion on the number of revolutions required in LES for the results to reach a steady-state is included.

### 2.1. MEXICO Experiment

The Mexnext Phase-III Model Experiments In Controlled Conditions (MEXICO) project was an international collaboration led by the Energy Research Centre of the Netherlands (ECN). The goal of the project was to generate big datasets that can be used to improve and develop the new aerodynamic models of wind turbines [3]. In this study, results of the (New) MEXICO experiment—which was made available in 2018—are used, which is a follow-up of Phase 1 and Phase 2 MEXICO experiments and resolves discrepancies of the results obtained for loads and velocities in Phase-1. The diameter of the MEXICO rotor blade is 4.5 m and consists of 3 airfoils viz. the DU91-W2-250 airfoil, RISØ-A2-21 airfoil, and NACA 64-418 airfoil from 20 to 45.6% span, 54.4% to 65.6% span, and from 74.4% till the tip of the blade, respectively, as shown in Figure 1. The MEXICO rotor is chosen here for the comparison of uRANS and LES turbulence modeling because along with the rotor measurements, it also includes detailed measurement data of the flow-field, unlike the NREL Phase VI experiment which only includes rotor measurements. The pressure measurement is obtained using 148 Kulite pressure sensors, distributed over 5 radial locations of $\frac{r}{R}$ = 0.25, 0.35, 0.60, 0.82, 0.92, and velocity components are measured using stereo PIV measurement technique. The dotted lines in Figure 1 indicate the locations of pressure measurements. The blade is twisted and tapered throughout its length, and the global pitch angle is kept at $-2.3°$. The blade is rotated at 425.1 rpm, and the experiments were performed in a wind tunnel for the wind speeds of 10, 15, and 24 ms$^{-1}$ which corresponds to turbulent wake state, design condition, and stall condition of the blade, respectively. In this study, the flow is perpendicular to the plane of rotation, and the effect of blade yaw is not considered.

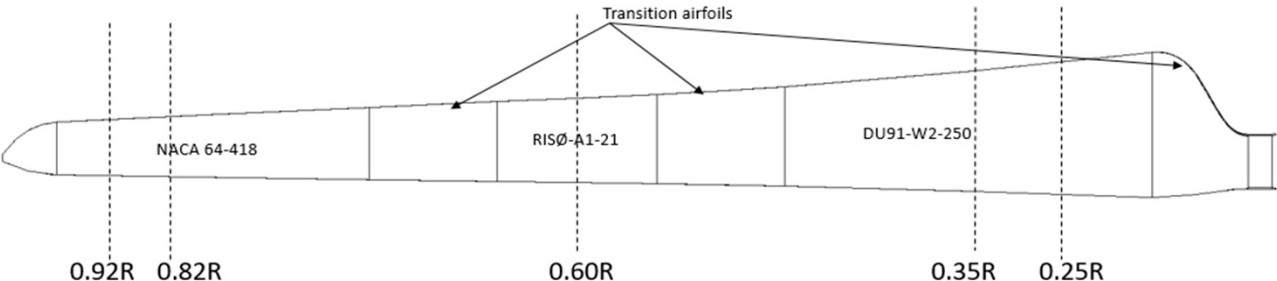

**Figure 1.** MEXICO rotor blade with airfoil distribution at different sections of the blade. The dotted vertical line represents the pressure measurement locations.

### 2.2. Computational Domain, Boundary Conditions, and Solution Methodology

The computational domain used for the CFD analysis is divided into 3 domains, as shown in Figures 2 and 3. The blade is enclosed in a small cylindrical domain (called the rotating domain) which is given a rotation of 425.1 rpm. A fine mesh domain is created in the region adjacent to the rotating domain, called stationary domain (fine), to capture near wake characteristics more accurately. These two domains are enclosed in a stationary domain, as depicted in Figure 2. The rotating domain is placed at the center of the stationary domain. The other dimensions of the computational domain are shown in Figure 2. The effect of the tower is not considered in the analysis. To model the relative motion between

the rotating domain and the stationary domain, a sliding mesh approach is used. This approach, however, is more computationally demanding than the other commonly used technique, called the Multiple Reference Frame (MRF) approach, but is more accurate and describes the transient phenomenon accurately [27].

The inlet is given a uniform velocity and 0 Pa gauge pressure is specified as pressure outlet boundary condition. The far-field surfaces are modeled with a symmetrical boundary condition. Turbulence is generally not high in the wind tunnels and the experiment does not provide any turbulence intensity value, keeping this in mind, a low value of 0.1% is chosen as turbulence intensity value and viscosity ratio $\left(\frac{\mu_t}{\mu}\right)$ is set as 10. Residual for continuity was kept as $1 \times 10^{-3}$ and all other variables were kept as $1 \times 10^{-5}$.

The unsteady RANS simulations were initially run at higher time steps to allow the flow to reach the outlet. The converged solution was then run at a low value of time-step corresponding to 1° rotation of blade per time-step is used ($3.92 \times 10^{-4}$ s) [17]. Running LES from scratch is very computationally demanding, therefore, to save computational time, the LES uses the initial converged solution of uRANS at a larger time-step. The RANS model is then changed to the WALE-LES model and a lower time-step value ($3.92 \times 10^{-4}$ s) is used for 4 revolutions of the blade.

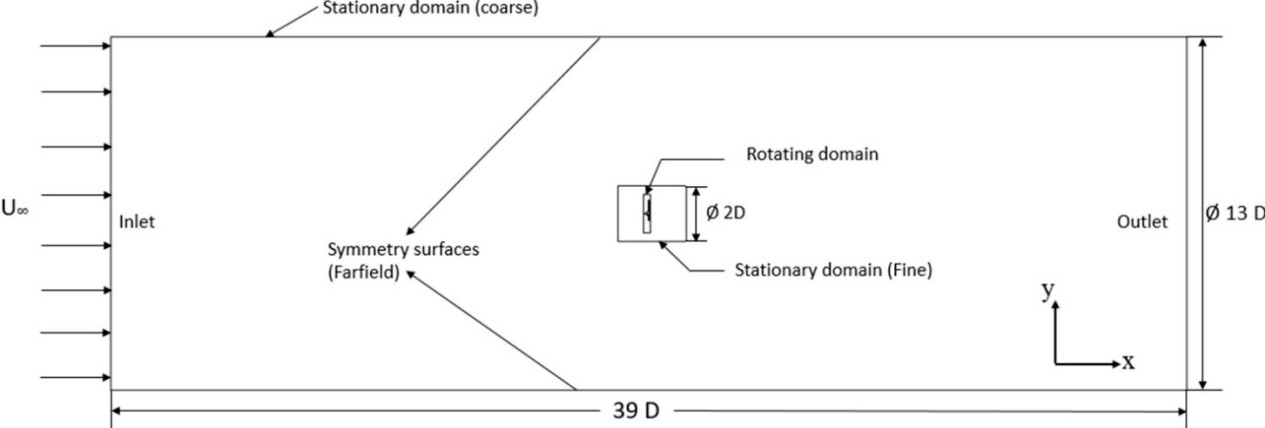

**Figure 2.** Entire computational domain used for the CFD analysis.

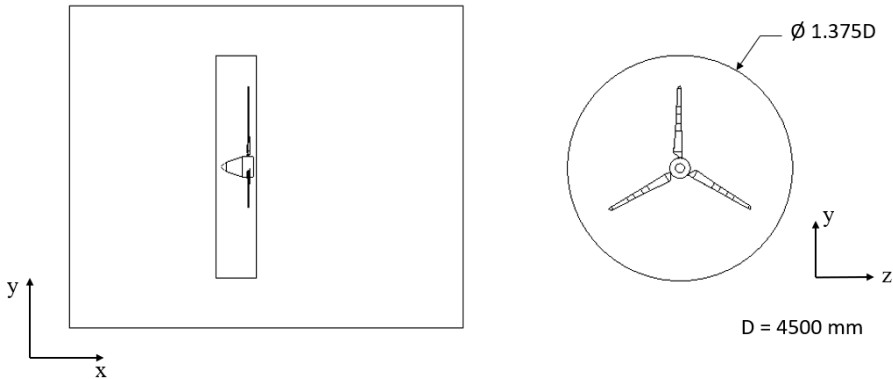

**Figure 3.** Zoomed view of the rotating domain and stationary domain (fine): (**left**) rotating domain inside a stationary domain of fine mesh (**right**) blade inside a rotating domain.

The ANSYS Fluent software (version 20.1) is used to perform the uRANS and LES simulations. Double-precision-based segregated solvers were applied. Pressure-velocity coupling is obtained with the SIMPLE algorithm. For better accuracy, the momentum and turbulence equations are spatially discretized using the second-order upwind scheme in uRANS. The momentum equations in the LES are discretized using a bounded central differencing approach. The Least Squares Cell-Based method is used to compute the

gradients in both approaches. A first-order temporal discretization is adopted for uRANS, whereas a second-order bounded implicit temporal discretization is used for the LES approach [21].

### 2.3. Mesh Details

A good quality unstructured mesh composed of tetrahedral and prism elements was generated using Ansys ICEM-CFD software. The mesh of the blade with a zoomed view of the mesh at the tip and root region is shown in Figure 4. A sufficiently fine mesh was created on the blade with 15 inflation layers with the initial height of $1 \times 10^{-5}$ m which gives y+ < 1, to resolve the boundary layer effects. The growth rate of 1.2 is used to extrude the inflation layer at the blade surface. Figure 5 shows zoomed view of the inflation layer mesh at the root region of the blade. The rotating and stationary domains were meshed independently and connected via mesh interfaces in Ansys FLUENT. The mesh consists of a total of 38.7 million elements, of which the rotating domain consisted of nearly 19 million elements and the rest of the elements were in the stationary domains.

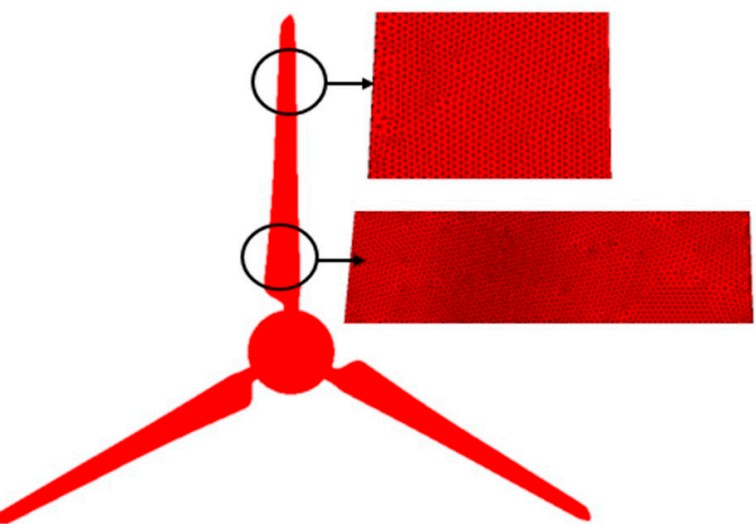

**Figure 4.** Mesh on the blade with a zoomed view of the mesh near the tip and hub region.

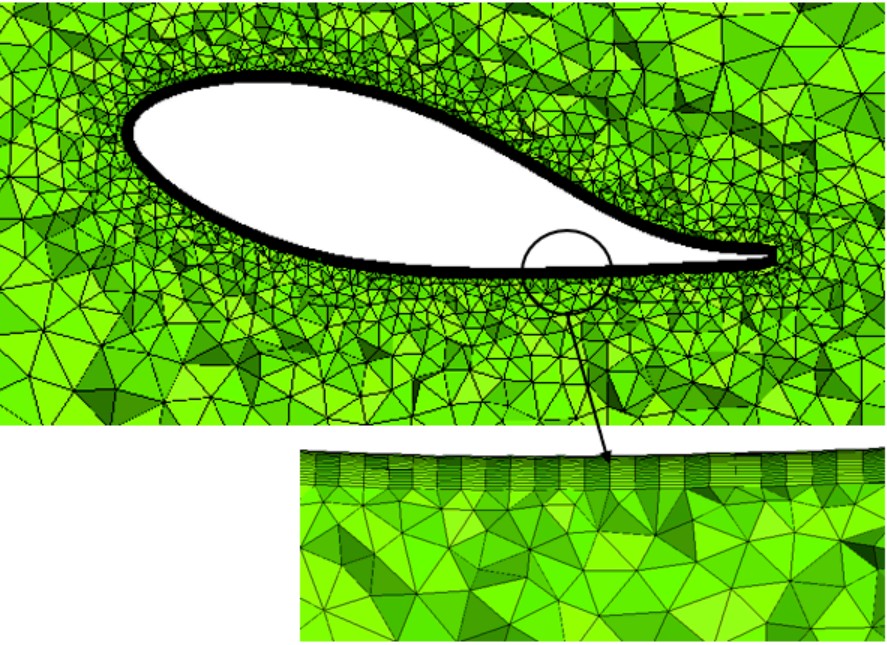

**Figure 5.** Zoomed view of the inflation layer in the root region of the blade.

### 2.4. Governing Equations

The flow fields in the large part of wind turbines are usually incompressible. Therefore, continuity and momentum equations can be written, respectively, as follows:

$$\frac{\partial}{\partial x_i}(u_i) = 0 \tag{1}$$

$$\rho\frac{\partial}{\partial t}(u_i) + \frac{\partial}{\partial x_j}\left(\rho u_i u_j + p\delta_{ij} - \tau_{ij}\right) = \rho f_i' \tag{2}$$

In the above equations, $\rho$ is the air density, $u_i$ and $u_j$ are the $i$th and $j$th direction velocity component (ms$^{-1}$), $p$ is the pressure (Pa), $f_i'$ is the body force per unit mass (N/kg), $\delta_{ij}$ is the Kronecker delta, and $\tau_{ij}$ are the stresses acting on the fluid particle (N/m$^2$). Here, $i$, and $j$ take values of 1, 2, 3 corresponding to $x$, $y$, and $z$-direction, respectively.

Turbulence models in CFD are methods for modeling the evolution of turbulence in fluid flows. Turbulent flows are commonplace in nature, and they occur in the majority of engineering applications; therefore, a turbulence model is required for the majority of simulations. Turbulence models of various sorts, such as RANS, DES, and LES, all have strengths and limitations owing to the nature of the modeling assumptions. The turbulence modeling chosen for this study is uRANS and LES, which are based on transient analysis and are widely considered to be more accurate than steady analysis. There are several types of closure models for RANS, with $k$-$\omega$ SST being the most commonly used for wind turbine CFD study. Similarly, among the various LES models, the Wall-Adaptive Local Eddy-Viscosity (WALE) model was chosen for comparison. The equations involved in both of these closure models are described next.

#### 2.4.1. uRANS $k$-$\omega$ SST Model

In this study, the closure model for uRANS equations is chosen as the $k$-$\omega$ SST turbulence model. It is a two-equation eddy-viscosity model that makes use of $k$-$\omega$ formulation in the boundary layer and switches to $k$-$\varepsilon$ in the free-stream. Among several eddy-viscosity turbulence models, the previous studies for flow through HAWTs [7,8,12,17,22,23,32,33] have shown that the $k$-$\omega$ SST model performs better for the flow simulations through a wind turbine. This model is especially reliable in adverse pressure gradients and separating flows—the two common phenomena encountered in wind turbines. The equations associated with the $k$-$\omega$ SST turbulence model can be described as follows:

$$\frac{\partial}{\partial t}(\rho k) + \frac{\partial}{\partial x_i}(\rho k u_i) = \frac{\partial}{\partial x_j}\left(\Gamma_k\frac{\partial k}{\partial x_j}\right) + \widetilde{G}_k - Y_k + S_k \tag{3}$$

$$\frac{\partial}{\partial t}(\rho\omega) + \frac{\partial}{\partial x_i}(\rho\omega u_i) = \frac{\partial}{\partial x_j}\left(\Gamma_\omega\frac{\partial\omega}{\partial x_j}\right) + G_\omega - Y_\omega + D_\omega + S_\omega \tag{4}$$

where $k$ is the turbulence kinetic energy (m$^2$/s$^2$); $\omega$ is the specific dissipation rate (s$^{-1}$), $Y_k$ and $Y_\omega$ are the dissipation rate of $k$ and $\omega$, respectively, $\Gamma_k$ and $\Gamma_\omega$ are the effective diffusivity of $k$ and $\omega$, respectively; $S_k$ and $S_\omega$ are the user-defined source terms of $k$ and $\omega$, respectively, $D_\omega$ is the cross-diffusion term, $G_\omega$ is the specific dissipation rate generation.

The reader is requested to refer to References [34,35] for the detailed description of the model and the equations involved.

#### 2.4.2. WALE LES Model

The turbulent flows are characterized by both large and small eddies spanning a wide spectrum of time and length scales. The main principle behind LES is that it resolves larger eddies and models smaller eddies using Sub Grid-Scale (SGS) modeling. It does so by using a spatial and temporal filtering operation on the Navier-Stokes Equations (1) and (2). Filter width and grid size are the two important parameters that influence the filtering

process. If the scale of the eddies is smaller than either of these two parameters, then such eddies get filtered out. In this way, it lies between RANS and DNS approaches in terms of the fraction of the resolved eddies.

The filtered Navier-Stokes equations can be written as:

$$\frac{\partial \rho}{\partial t} + \frac{\partial}{\partial x_i}(\rho \overline{u_i}) = 0 \tag{5}$$

$$\frac{\partial}{\partial t}(\rho \overline{u_i}) + \frac{\partial}{\partial x_j}(\rho \overline{u_i u_j}) = \frac{\partial}{\partial x_j}(\sigma_{ij}) - \frac{\partial \overline{p}}{\partial x_i} - \frac{\partial \tau_{ij}}{\partial x_j} \tag{6}$$

The stress tensor due to molecular viscosity, $\sigma_{ij}$, can be defined as:

$$\sigma_{ij} \equiv \left[\mu \left(\frac{\partial \overline{u_i}}{\partial x_j} + \frac{\partial \overline{u_j}}{\partial x_i}\right)\right] - \frac{2}{3}\mu \frac{\partial \overline{u_l}}{\partial x_l} \delta_{ij} \tag{7}$$

And the subgrid-scale stress, $\tau_{ij}$, is defined as:

$$\tau_{ij} = \rho \overline{u_i u_j} - \rho \overline{u_i} \overline{u_j} \tag{8}$$

The above subgrid-scale stresses brought by the application of filtering operation are not known a priori and therefore must be modeled. The subgrid-scale turbulence stress can be obtained from:

$$\tau_{ij} - \frac{1}{3}\tau_{kk}\delta_{ij} = -2\mu_t \overline{S_{ij}} \tag{9}$$

The WALE model is chosen to obtain eddy-viscosity, $\mu_t$, because it does not require any wall or damping functions and its algebraic nature performs simulations faster.

$$\mu_t = \rho L_s^2 \frac{\left(S_{ij}^d S_{ij}^d\right)^{3/2}}{\left(\overline{S}_{ij}\overline{S}_{ij}\right)^{5/2} + \left(S_{ij}^d S_{ij}^d\right)^{5/4}} \tag{10}$$

where, $S_{ij}$ and $\overline{S}_{ij}$ are square of velocity gradient tensor and shear stress tensor, respectively. The WALE models, $L_s$ and $S_{ij}^d$, are defined, respectively, as:

$$L_s = min\left(\kappa d, \, C_w V^{1/3}\right) \tag{11}$$

$$S_{ij}^d = \frac{1}{2}\left(\overline{g}_{ij}^2 + \overline{g}_{ji}^2\right) - \frac{1}{3}\delta_{ij}\overline{g}_{kk}^2 \tag{12}$$

In the above equations, $\kappa$ is the von Karman constant, $d$ is the distance to the closest wall (m), $V$ is the computational cell volume (m³), $\overline{g}$ is the velocity gradient tensor The default value of WALE constant, $C_w$, is 0.325.

### 2.5. Number of Rotor Revolutions for Steady-State Solution in LES

It is a well-known fact that to achieve stability in the LES solutions, it needs to be run for sufficiently longer flow times than the RANS simulations. Therefore, a sensitivity study is presented here to test the number of blade revolutions needed to achieve steady-state solutions. Figure 6 shows a comparison of axial, radial, and tangential velocity plots with experiments, at different revolutions of the turbine blade, for the wind speed of $24 \, \text{ms}^{-1}$. The mean absolute deviation for all three components of velocity is less than 1% for the three different revolutions of the blade considered in the analysis. Keeping the computational cost associated with LES in mind, the simulations here are, thus, run for four revolutions of the turbine blade.

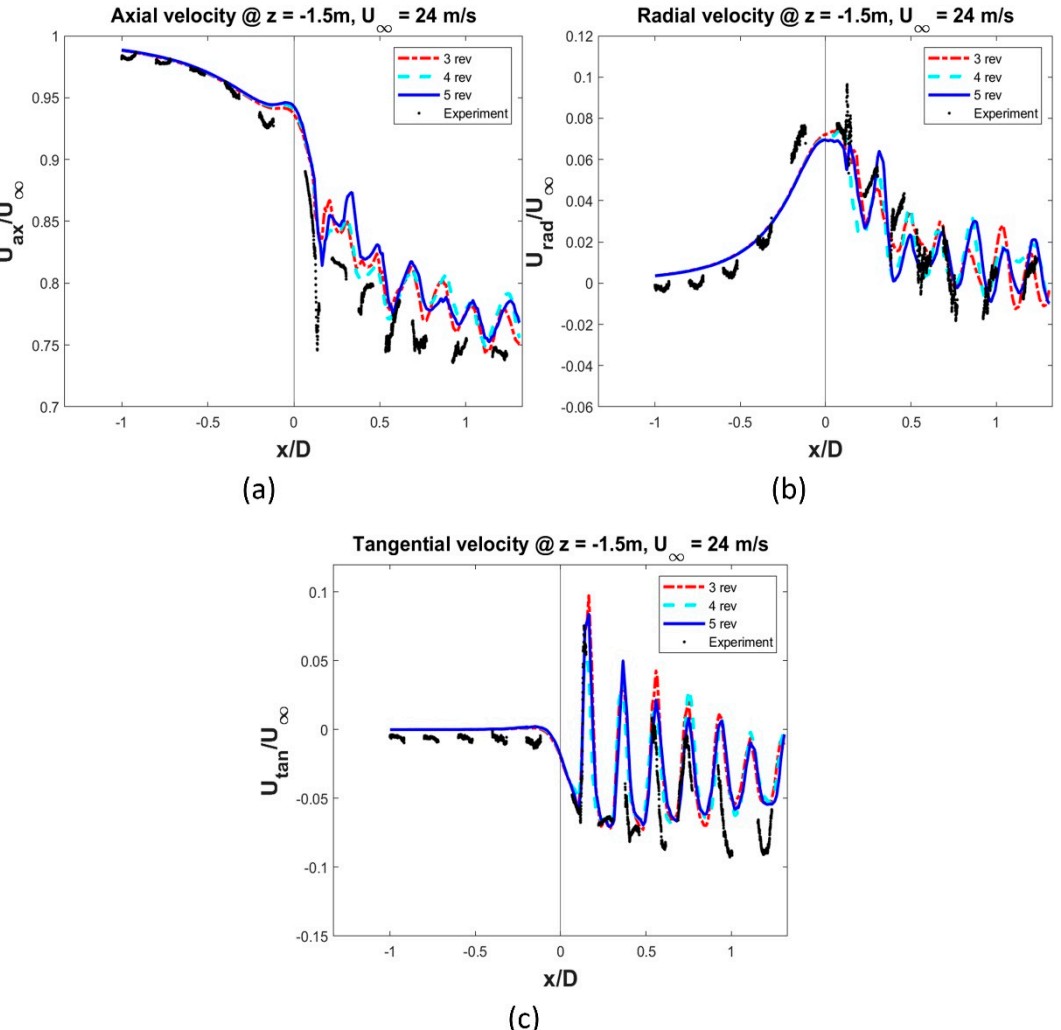

**Figure 6.** Investigation of the number of revolutions needed to achieve state solution in LES; (**a**) axial; (**b**) radial; (**c**) tangential velocity at z = −1.5 m for 24 ms$^{-1}$ wind speed.

## 3. Results and Discussions

### 3.1. Comparison of Pressure Distribution at 5 Radial Locations

The pressure coefficient ($C_p$) distributions at the 5 radial locations for the wind speeds of 10, 15, and 24 ms$^{-1}$ which corresponds to the Tip speed ratio ($\lambda$) = $\frac{\Omega R}{U_\infty}$ = 10, 6.67, and 4.17, respectively are shown in Figures 7–9. The $C_p$ equation can be given as:

$$C_p = \frac{P - P_\infty}{0.5\rho\left(U_\infty^2 + (\omega r)^2\right)} \tag{13}$$

The $C_p$ values predicted from the two modeling approaches, both on the suction and pressure surfaces, overlap each other for the tip speed ratio ($\lambda$) of 10, as shown in Figure 7. The $C_p$ distributions obtained from the CFD computations are in poor agreement in the inboard region of the blade, especially at the radial location of 0.25R, due to insufficient resolution of the pressure sensors to resolve the flow physics [3]. Similar $C_p$ distribution at the radial location of 0.25R was also obtained in previous works [6,8,13,17,22,32,33]. The agreement between CFD predictions and experimental measurements at the mid-span and outboard region is, however, excellent, as evident from Figure 7. The $C_p$ distribution predicted for the wind speed case of 15 ms$^{-1}$ by both models, as presented in Figure 8, is in good agreement with the experimental values, with slight differences on the suction side. The LES model underpredicts pressure in the leading-edge region at the suction

side in the inboard and mid-span region. On the other head, both models give nearly the same pressure prediction at the outboard region of the blade when the wind turbine operates in the design condition ($U_\infty$ = 15 ms$^{-1}$, $\lambda$ = 6.67). The differences in pressure distribution obtained from the two modeling approaches can be seen clearly for the wind speed of 24 ms$^{-1}$, as evident from Figure 9. At the wind speed of 24 ms$^{-1}$ ($\lambda$ = 4.167), the turbine operates in stalled flow conditions with massive flow separation. At such high-speed flows, enhancement in suction peak is observed and the slope of pressure gradient reduces. The separation is said to occur when the pressure gradient slope $\left(\frac{dP}{dx} = 0\right)$ does not change on the blade surface. This separation point is better predicted by the LES model as compared to the uRANS model, which predicts delay in the flow separation, as can be seen from Figure 9. The LES model overpredicts pressure distribution at the leading-edge region located at outboard stations (at spans of 0.82R and 0.92R). Overall, pressure distributions obtained from both models at different blade spans match well with the experimental measurements.

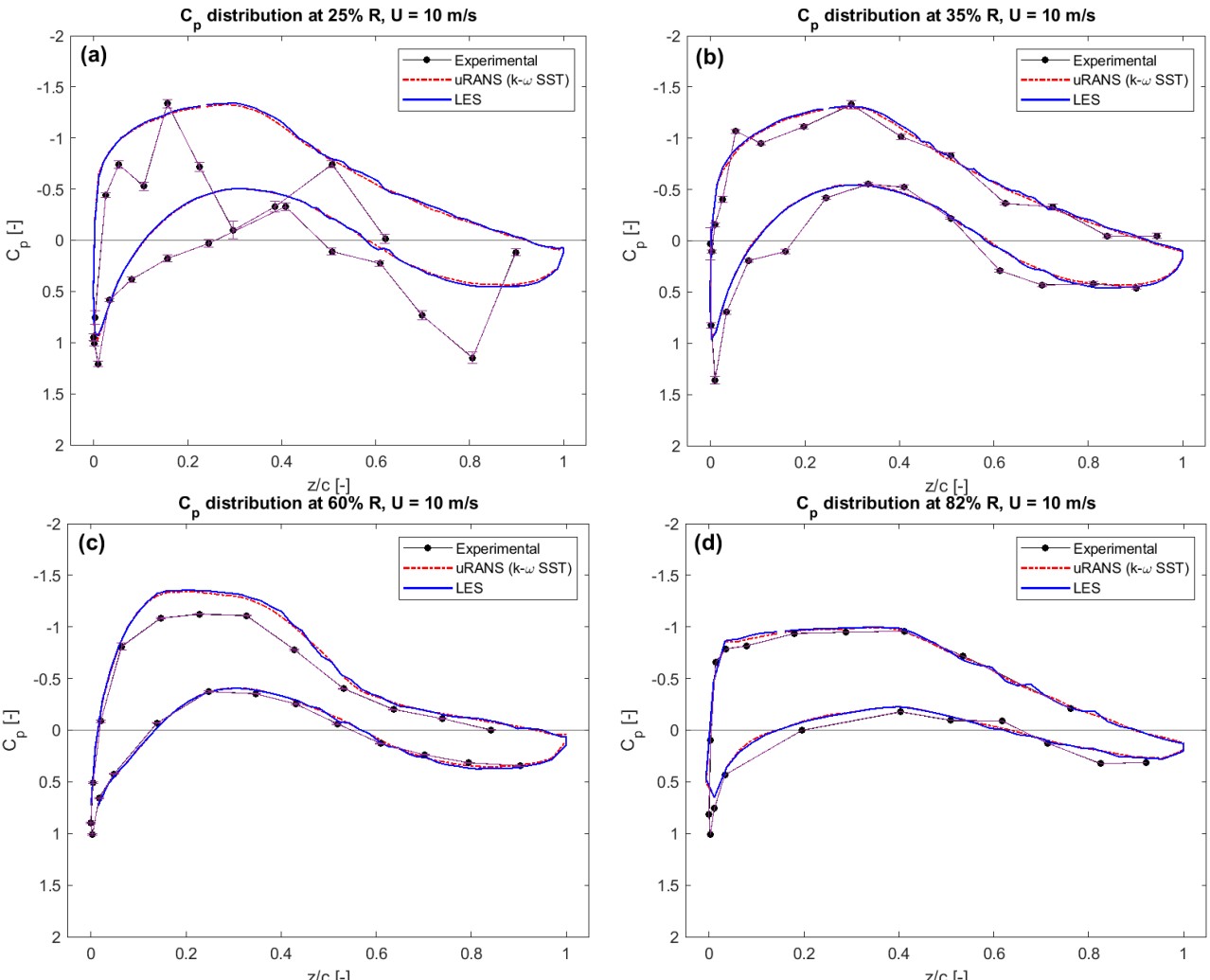

**Figure 7.** *Cont.*

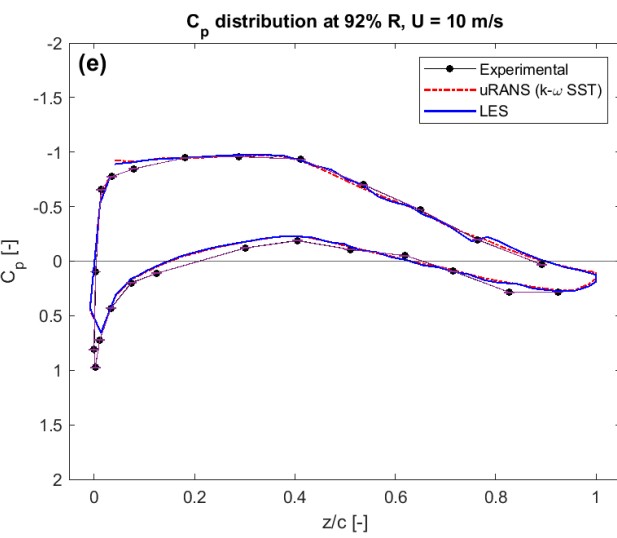

**Figure 7.** Pressure coefficient distribution at the 5 radial locations: $\frac{r}{R}$ = (**a**) 0.25; (**b**) 0.35; (**c**) 0.60; (**d**) 0.82, and (**e**) 0.92; for the freestream wind velocity (*U*) of 10 ms$^{-1}$ ($\lambda$ = 10).

**Figure 8.** *Cont.*

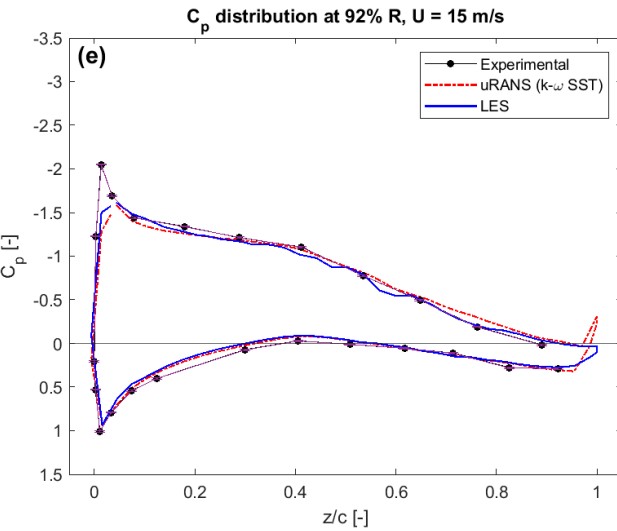

**Figure 8.** Pressure coefficient distribution at the 5 radial locations: $\frac{r}{R} =$ (**a**) 0.25; (**b**) 0.35; (**c**) 0.60; (**d**) 0.82, and (**e**) 0.92; for the freestream wind velocity (*U*) of 15 ms$^{-1}$ (λ = 6.67).

**Figure 9.** *Cont*.

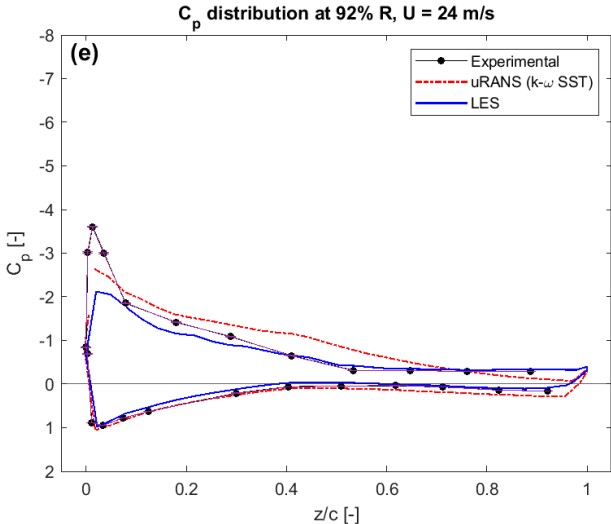

**Figure 9.** Pressure coefficient distribution at the 5 radial locations: $\frac{r}{R}$ = (**a**) 0.25; (**b**) 0.35; (**c**) 0.60; (**d**) 0.82, and (**e**) 0.92; for the freestream wind velocity (*U*) of 24 ms$^{-1}$ ($\lambda$ = 4.17).

### 3.2. Velocity Distributions

### 3.2.1. Axial Traverse

Figure 10 depicts the axial traverse conventions used in plotting the three components of the velocity. The axial, radial and tangential components of velocities are plotted at a distance of $z = -1.5$ m when the rotor is in the position shown in Figure 10a. The axial traverse covers the measurement range from $x = -4.5$ m (upstream) to $x = 5.9$ m (downstream). The corresponding velocity components plots for the wind speed of 15 and 24 ms$^{-1}$ are included in Figure 11. The vertical line at $x/D$ =0 represents the rotor plane.

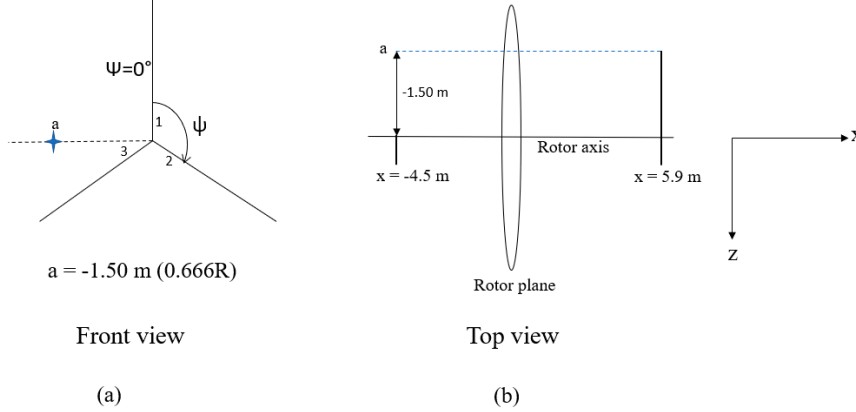

**Figure 10.** Axial traverse conventions and its definitions in the MEXICO rotor: (**a**) Front view; (**b**) Top view.

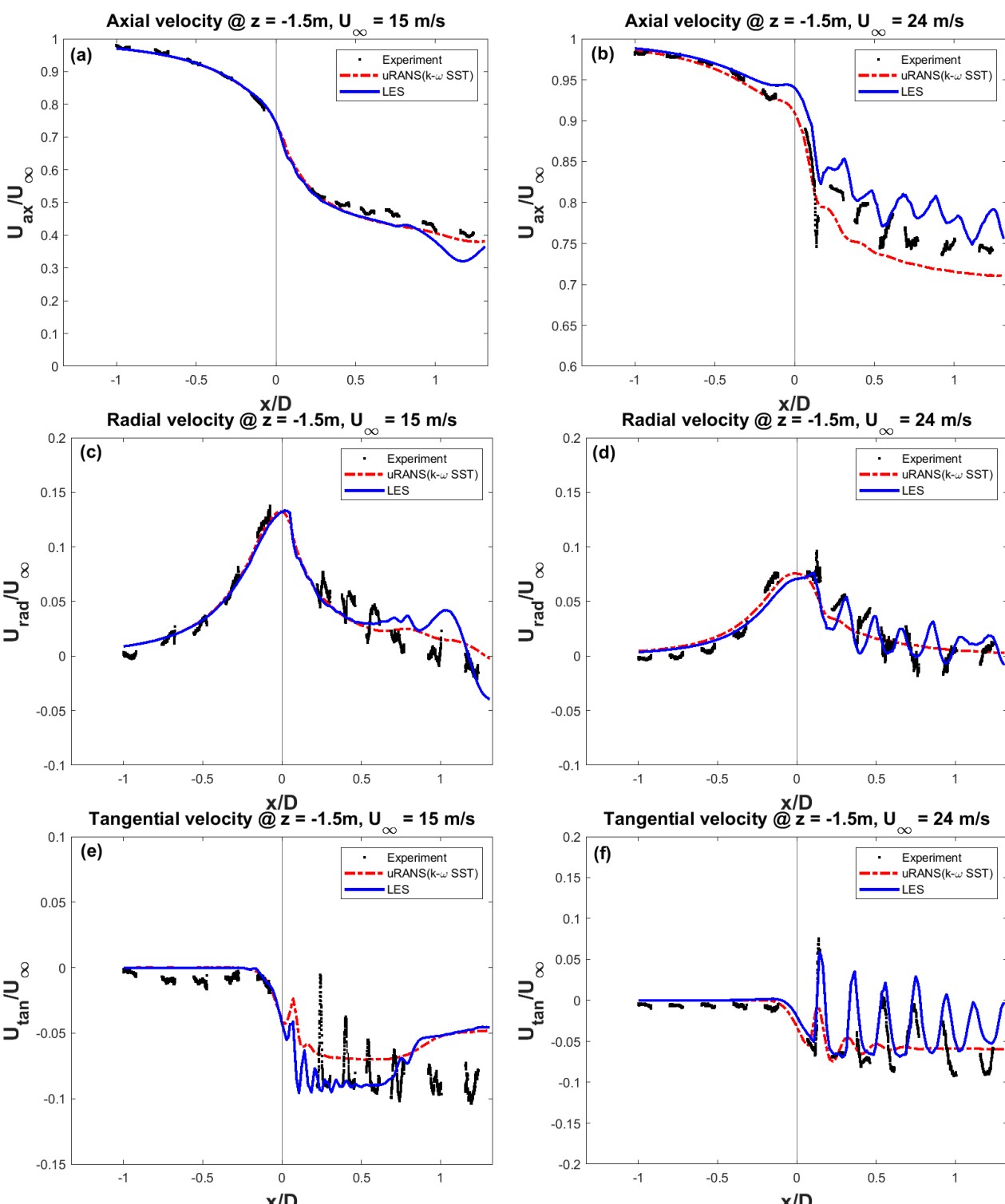

**Figure 11.** Axial traverse of 3 components of velocity at the location *z* = 1.5 m for the free-stream wind speed of 15 m/s and 24 m/s: (**a**,**b**) axial velocity; (**c**,**d**) radial velocity; (**e**,**f**) tangential velocity.

The three components of velocity i.e., axial, radial, tangential, give information about velocity deficit, wake expansion, and wake rotation, respectively. All the three components of velocity are predicted well by both modeling approaches, as can be seen in Figure 11a–f. The velocity components upstream of the blade are predicted accurately by both modeling approaches indicating the outer domain size chosen in the analysis is appropriate to exclude

the tunnel effects. The oscillations in the radial and tangential velocity for the wind speed case of 15 ms$^{-1}$ are not predicted accurately by both modeling approaches.

The differences in velocity components predicted by LES and uRANS can be seen much more clearly at the wind speed of 24 ms$^{-1}$ (Figure 11b,d,f). The undulation in the axial velocity due to flow separation on the blades (Figure 11b) is satisfactorily predicted by the LES approach. Due to unsteady wake effects, the oscillations in the radial and tangential velocity components (Figure 11d,f) behind the rotor plane are better captured by the LES approach. The uRANS approach, on the other hand, averages out such oscillatory patterns.

### 3.2.2. Radial Traverse

This section compares the radial traverse of velocity components obtained using the LES and uRANS modeling approaches, which are plotted downstream of the turbine blade at a distance of 0.3 m from the rotor plane as shown in Figure 12. The velocity components are plotted when the blade orientation is as indicated in Figure 12a, along a solid, vertical line in Figure 12b.

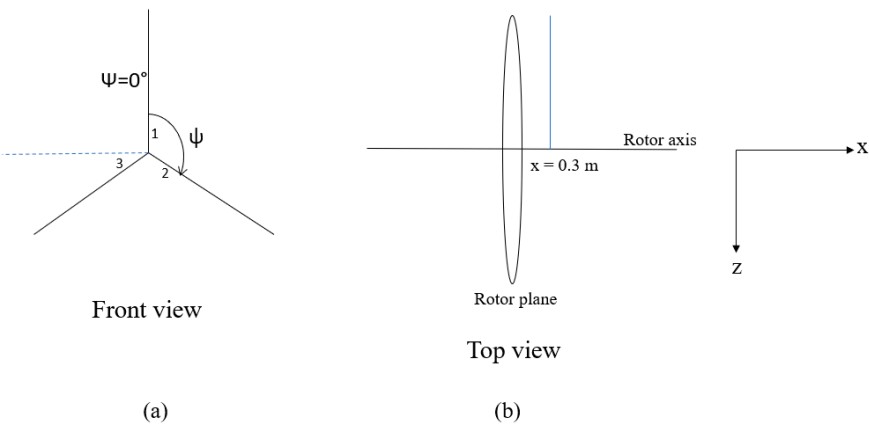

Front view              Rotor plane

                         Top view

(a)                               (b)

**Figure 12.** Radial traverse conventions and its definitions in the MEXICO rotor: (**a**) front view; (**b**) top view.

The axial component of velocity for the wind speed of 10 and 15 m/s (Figure 13a,d), are predicted slightly closer by LES than uRANS against the experimental measurements. For the wind speed of 24 m/s (Figure 13g), the LES model follows the experimental trend of velocity deficit and recovery from $0.2 < \frac{r}{R} < 0.4$ but shows faster wake recovery from $0.4 < \frac{r}{R} < 1$, than seen in the experiments. The uRANS, on the other hand, is unable to capture the wake deficit and recovery in the inboard region ($0.2 < \frac{r}{R} < 0.4$) but predicts much closer wake velocity at the mid-span and outboard region than the LES model, as evident from Figure 13g. The tip vortices, as can be seen from the sudden jump in axial velocity (Figure 13a,d) at r/R = 1, are predicted well by both modeling approaches for the wind speed of 10 and 15 ms$^{-1}$. Whereas, such tip vortices are not predicted well when the rotor operates at the wind speed of 24 ms$^{-1}$.

When the blade operates in stall condition ($\lambda$ = 4.17), the separated flow in the mid-span and outboard region is accurately predicted by both of these modeling approaches (Figure 13h,i). The rotation in the wake (tangential component of velocity) is predicted satisfactorily by both of these modeling approaches as can be seen from Figure 13c,f,i. The vortex shed by the blade tip is predicted more accurately for the wind speeds of 10 and 15 ms$^{-1}$ than for the wind speed of 24 ms$^{-1}$. Overall, the predicted trends in the radial traverse of velocity components are closer to the experimental measurements.

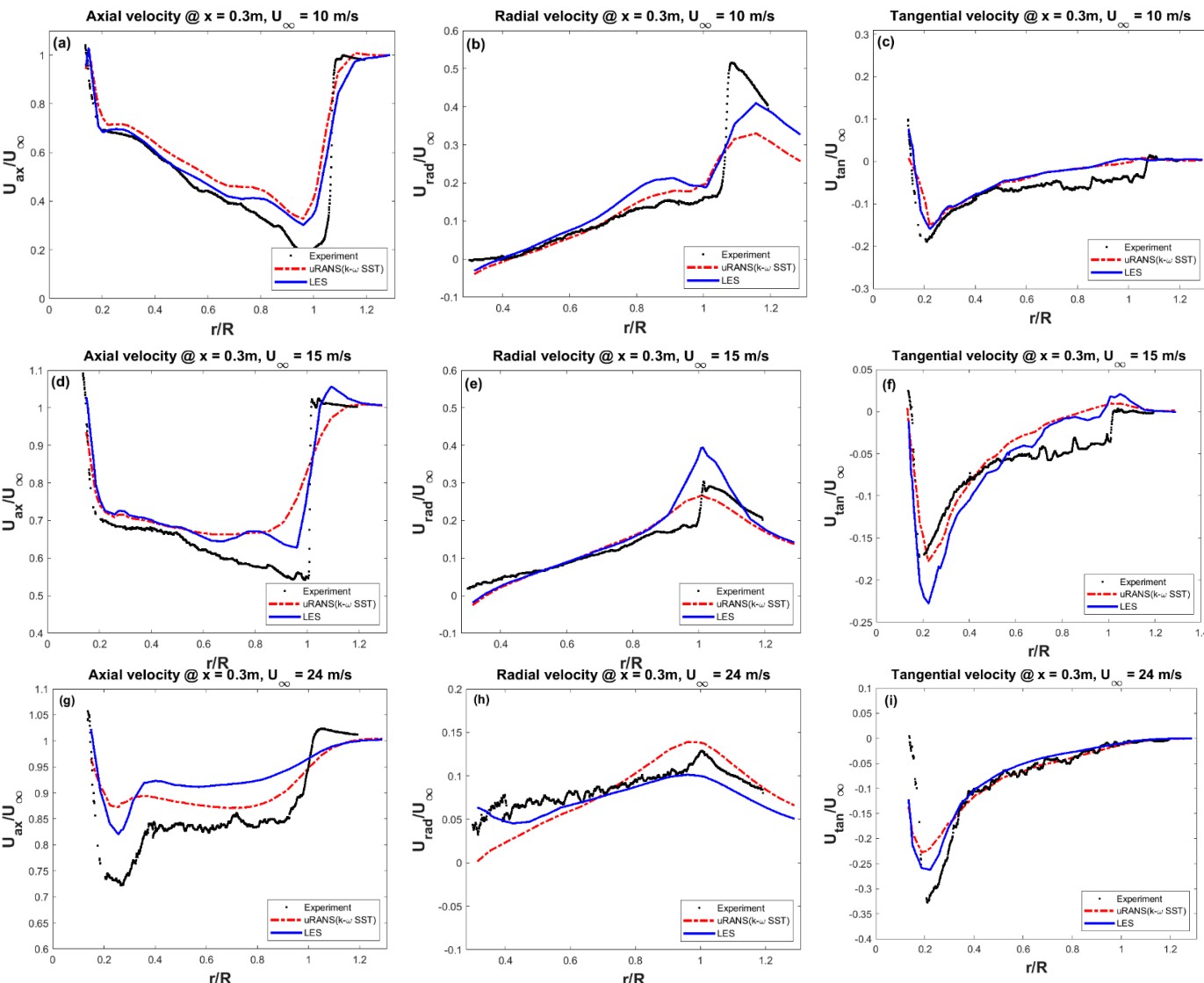

**Figure 13.** Radial traverse of 3 components of velocity at the location *x* = 0.3m for the free-stream wind speed of 10, 15, and 24 m/s: (**a**,**d**,**g**) axial velocity; (**b**,**e**,**h**) radial velocity; (**c**,**f**,**i**) tangential velocity.

### 3.2.3. Azimuthal Traverse

Figure 14 shows the conventions and definitions of azimuthal traverse of velocity components. Figures 15–17 show a comparison of the azimuthal traverse of the three components of velocity (axial, radial, and tangential) with the experimental measurements for the λ of 10, 6.67, and 4.17, respectively. The figures are plotted at a distance of 0.3 m downstream from a rotor plane (Figure 14b) at five radial locations $\frac{r}{R}$ = 0.25, 0.35, 0.60, 0.82, and 0.92 (Figure 14a). Radial velocity can be defined as the component of wind velocity directed away from the rotor center, perpendicular to the axial direction, whereas tangential velocity is the wind component aligned along the rotor rotational direction [36]. Comparing the axial velocity plot (Figure 15a), it can be observed that the prediction is closer to the measurements at the inboard and mid-section of the blade. Slight deviation can be seen in the outboard region (r = 0.82R and 0.92R), where the prediction of the axial velocity component is closer to the measured value than the uRANS approach. Overall, the trend in axial velocity components of both the upwash and downwash of the blade is well captured by uRANS and LES. The axial velocity is, however, overpredicted by both uRANS and LES at the outboard region. The vortex shed by the inboard region (r = 0.25R and 0.35R) is not captured well in the downwash of the blade. The radial and

the tangential components of velocity during upwash of the blade are not captured well by both of these approaches. Due to the separated blade wake and the root vortices in the inboard region for the lower wind speed case (10 ms$^{-1}$), this region is difficult to predict. At the outboard region, radial component prediction obtained using uRANS is closer to the measurements, whereas LES overpredicts the radial velocity component. The tangential velocity predicted by both of these approaches nearly overlaps each other for all the radial positions considered (Figure 15c).

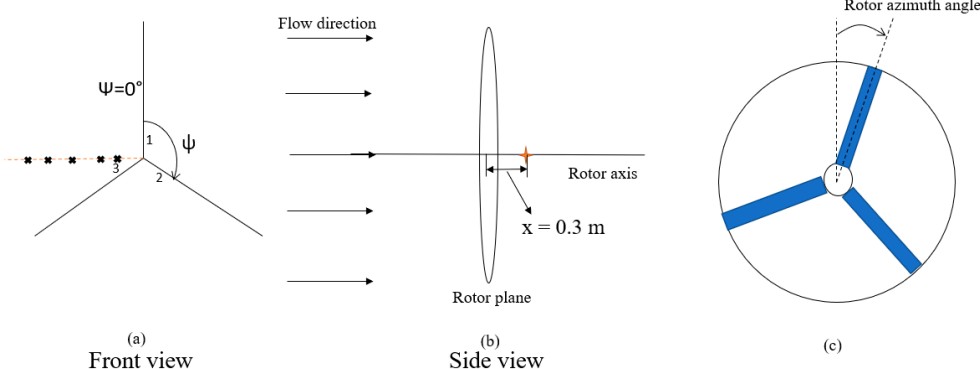

**Figure 14.** Azimuthal traverse conventions and its definitions in the MEXICO rotor.

An excellent agreement between numerical and experimental measurements of the three velocity components of the azimuthal traverse is observed when the blade operates at the design condition (i.e., $\lambda$ = 6.67), as can be seen in Figure 16. Compared to the turbulent wake state azimuthal plots, the tip and root vortices from the blade are better captured at this condition. The only appreciable difference in the axial velocity is observed at the outermost radial location of the blade (r = 0.92R), where the LES computations are closer to the experimental measurements than the uRANS predictions. The mean absolute error *(MAE)* in axial velocity computations and experimental measurements at r = 0.92R for all azimuthal positions is 18.40% and 12.04% from the uRANS and LES approaches, respectively. The *MAE* can be defined as:

$$MAE = \left| \frac{X_{CFD} - X_{exp}}{X_{exp}} \right| \times 100\% \tag{14}$$

Similarly, the radial velocity component (Figure 16b) is also approximated well by both modeling approaches with marginal deviations from the measurements. The tangential velocity computations from the LES approach are closer to the experimental measurements than the uRANS approach, with the average error being 18.5% and 10% from uRANS and LES, respectively. Moreover, as evident from Figure 16c, the tangential velocity during the downwash of the blade is better predicted by the LES approach than the uRANS approach.

Finally, a comparison of azimuthal traverse for the lowest tip speed ratio ($\lambda$ = 4.17) corresponding to the wind speed of 24 ms$^{-1}$ is shown in Figure 17. The axial velocity component (left side of Figure 17) in the inboard region is predicted well by the LES approach, whereas agreement is better between measurements and uRANS computations at the outboard region. A good agreement is also observed for the radial component of velocity as well, as evident from the plots in Figure 17b. The jump in the tangential velocity at 70° rotor azimuth due to induction effects, is captured well by the LES method. Overall, both approaches follow the general trend of the experimental observations.

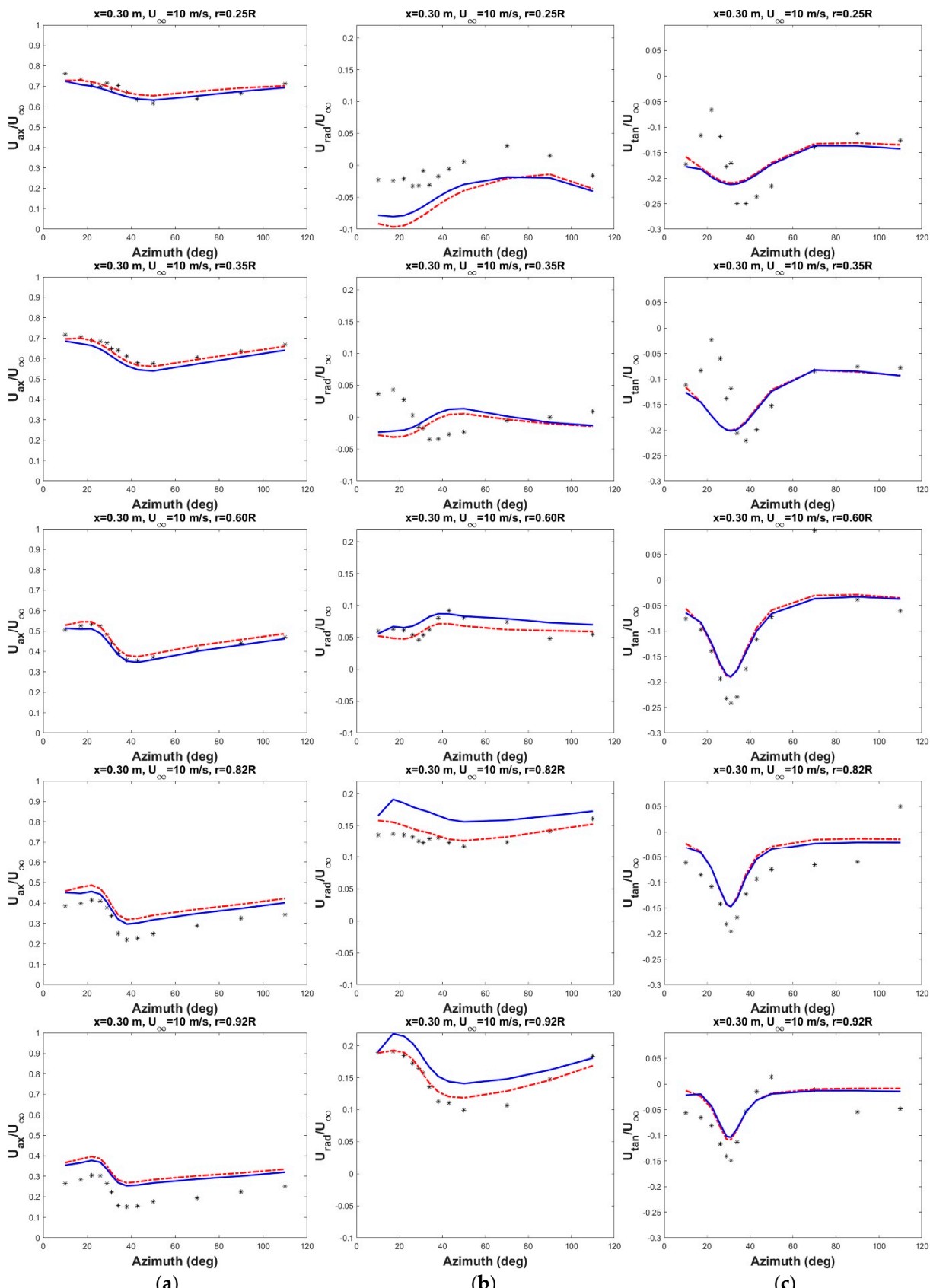

**Figure 15.** Azimuthal traverse of (**a**) axial; (**b**) radial; (**c**) tangential velocity for the wind speed of 10 ms$^{-1}$. Notations: * Experimental, – – – uRANS, ⎯⎯⎯ LES.

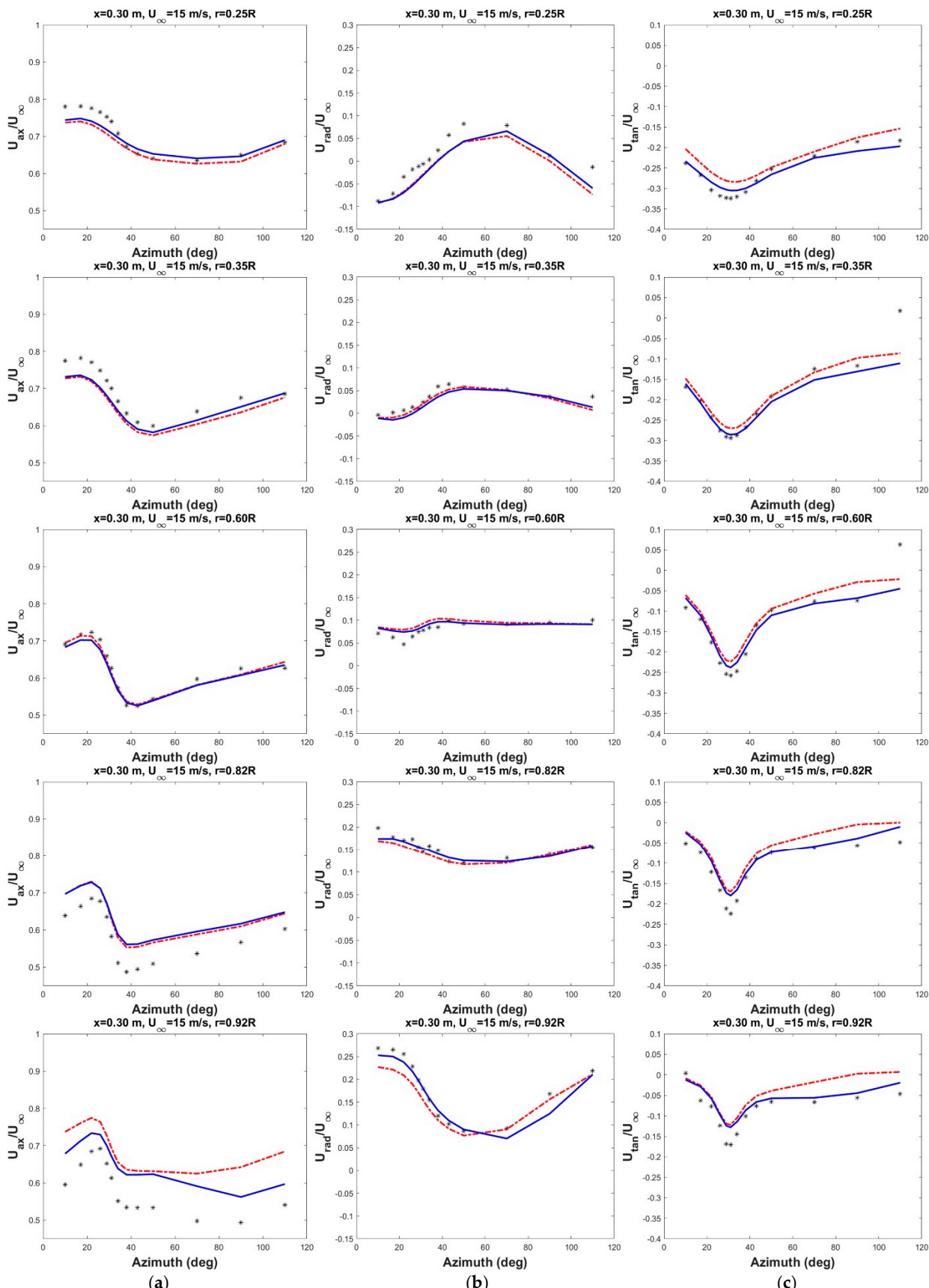

**Figure 16.** Azimuthal traverse of (**a**) axial; (**b**) radial; (**c**) tangential velocity for the wind speed of 15 ms$^{-1}$. Notations: * Experimental, – – – uRANS, ——— LES.

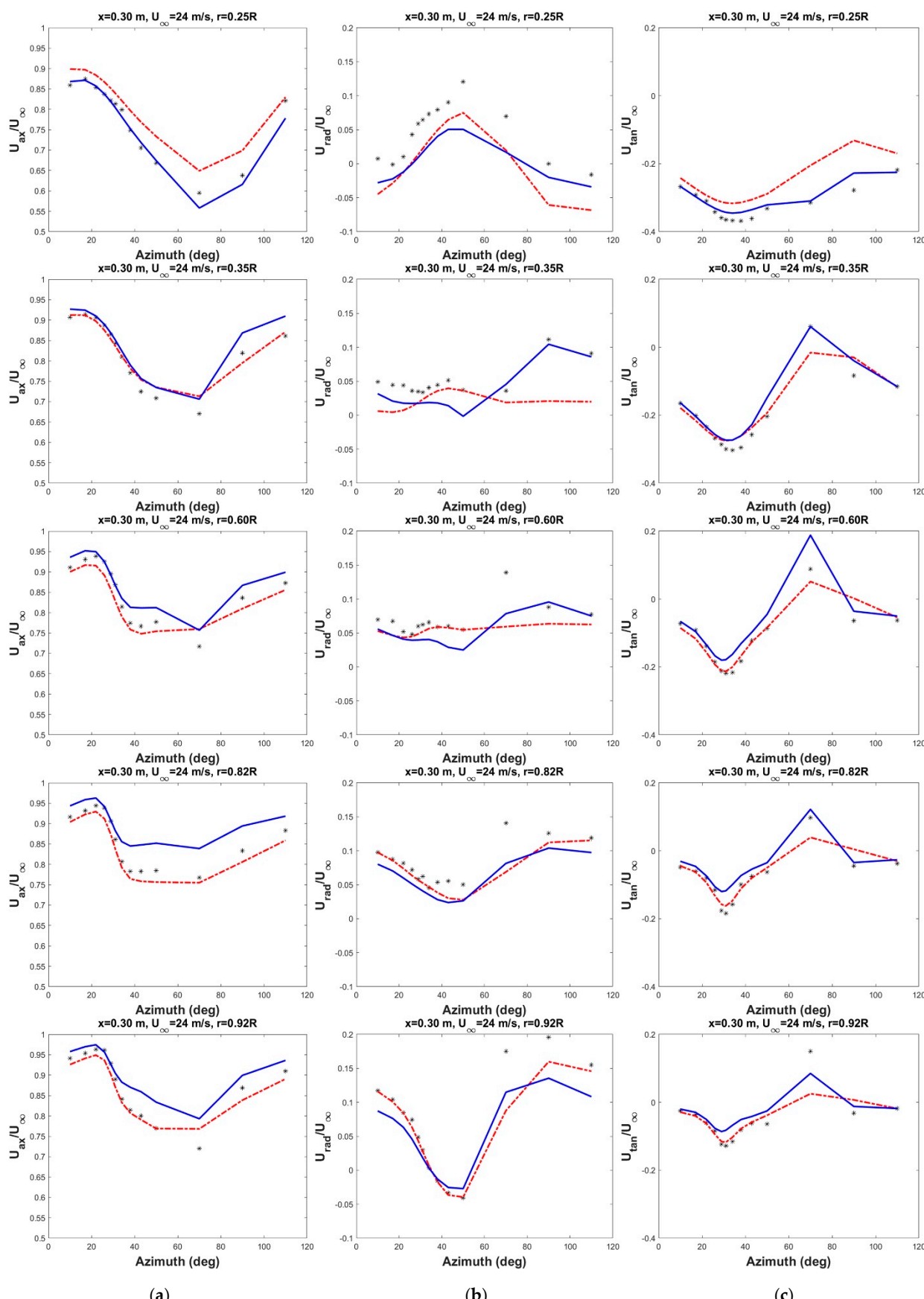

**Figure 17.** Azimuthal traverse of (**a**) axial; (**b**) radial; (**c**) tangential velocity for the wind speed of 15 ms$^{-1}$. Notations: * Experimental, – – – uRANS, —— LES.

### 3.3. Discussion on Near Wake Velocity at Several Downstream Locations

Insight into near wake is vital to improve wind farm efficiency as average wind farm losses due to wake could reach up to 20% of the total output of a large wind farm [37]. The near wake region can be defined as the downstream distance up to which blade influence can be experienced and after that distance, the wake profile is Gaussian in shape. Figure 18 represents velocity at various downstream locations obtained from uRANS and LES. It should be noted that wake radius increases with an increase in downstream distance. Therefore, to show the complete wake profile, r/R exceeds 1. Here, 'r' represents the local wake radius while 'R' is the blade radius. As mentioned, the overall wake radius might be greater than the rotor radius, therefore, we have included r/R greater than 1. The point where the velocity ratio ($U_{wake}/U_\infty$) reaches 1 is the free-stream region and the corresponding 'r' is the wake radius. The wake recovery for the wind speed of 10 ms$^{-1}$ is found to be similar from both uRANS and LES approaches. The wake is recovered faster by the uRANS approach than LES for the wind speeds of 15 and 24 ms$^{-1}$, as evident from Figure 18c,d. The curves are symmetric around the rotor axis because the effect of the tower is not considered in the analysis. Due to intense wake mixing at higher wind speeds, velocity deficit is lower at higher TSR, and vice-versa. The wake profile is Gaussian for all the tip-speed ratios after 3 rotor diameters downstream. Similar trends as shown in Figure 18d were also observed at farther downstream locations.

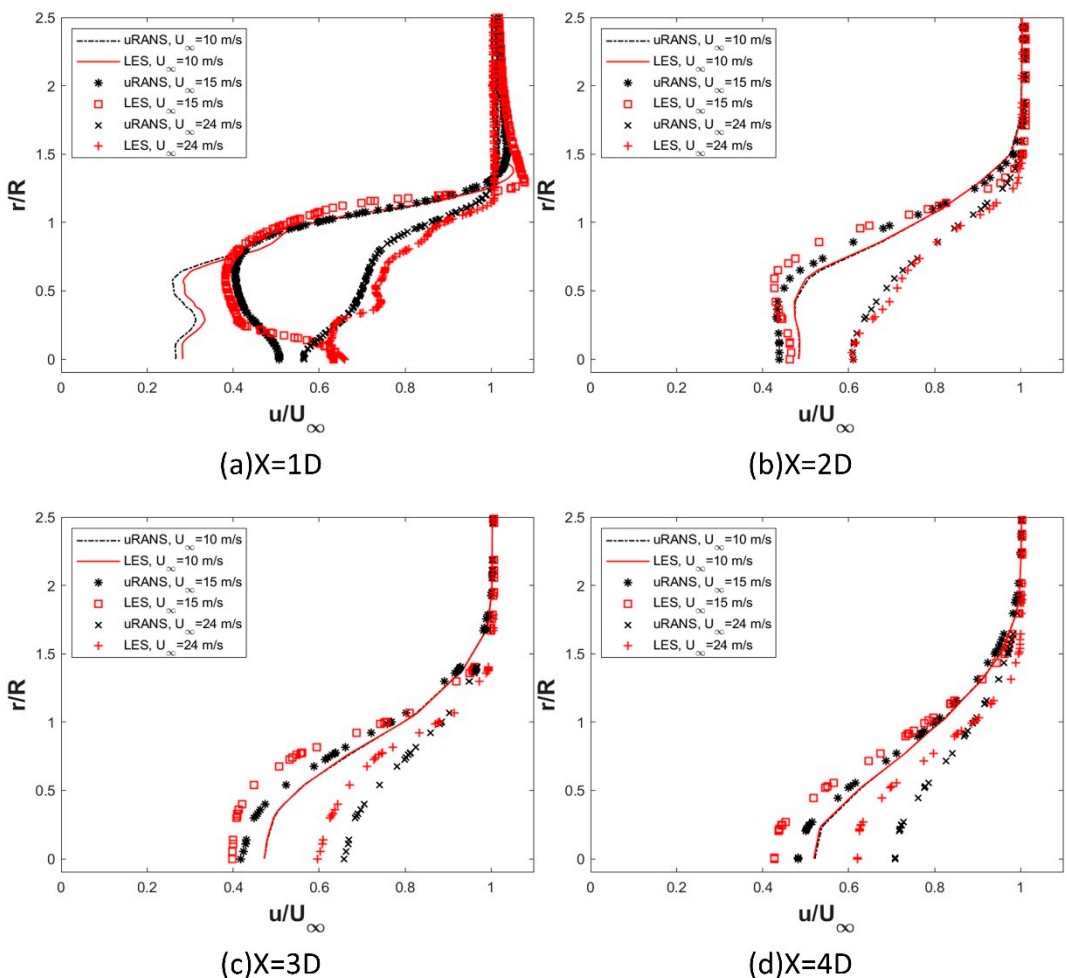

**Figure 18.** Wake velocities at various downstream locations of the wind turbine.

### 3.4. Discussion on Vorticity

Figure 19 shows the non-dimensional vorticity contours, given by the expression:

$$\zeta_n = \frac{\zeta D}{u_{ref}} \tag{15}$$

where $\zeta$ is the local vorticity magnitude (in s$^{-1}$), $D$ is the rotor diameter, and $u_{ref}$ is the hub height velocity. From Figure 19, it can be observed that near wake vortices are resolved much better by LES than uRANS, as evident from the small vortical turbulence structures resolved by the LES approach. Such structures are averaged out in the uRANS approach.

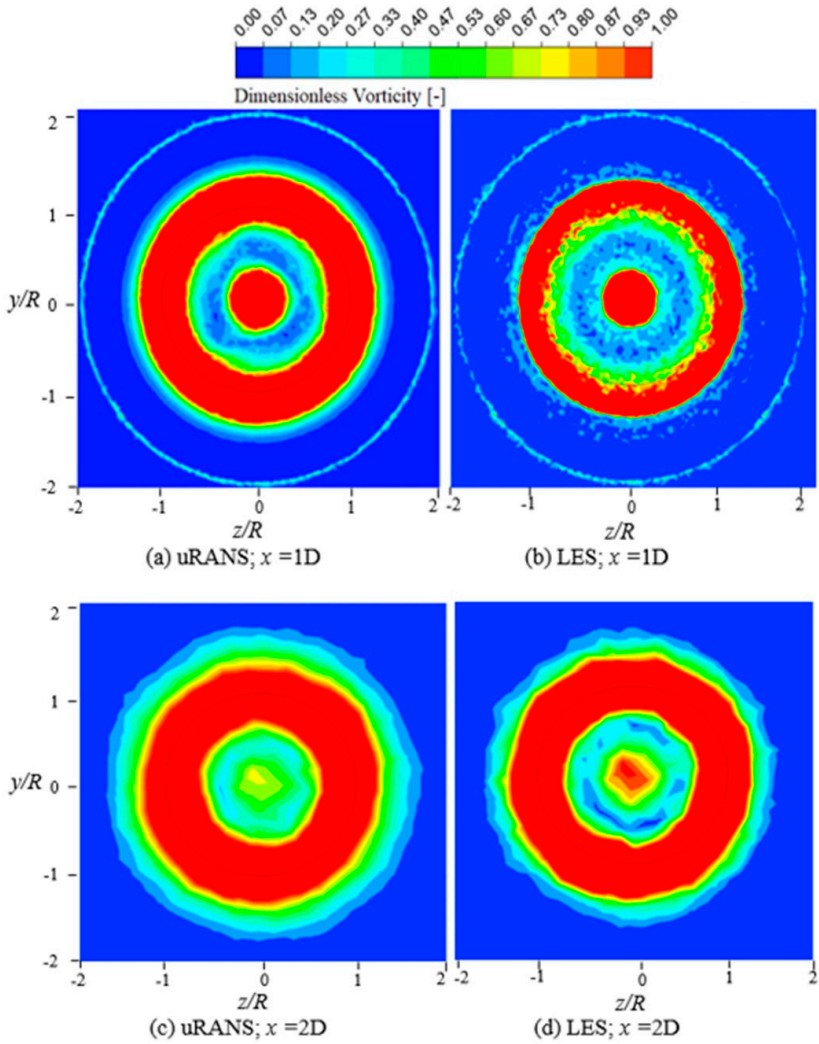

**Figure 19.** Vorticity contours for $\lambda$ = 6.67 at two downstream locations of $x$ = 1D (**top**) and $x$ = 2D (**bottom**).

Vortex is identified by using the $\lambda_2$ criterion developed by Jeong and Hussain [38]. To discern the presence of the vortex, they make use of pressure minimum definition by eliminating the effects of unsteady straining and viscous effects. Mathematically, pressure minimum created by vortex motion is defined as $S^2 + \Omega^2$, where $S$ and $\Omega$ are the symmetric and anti-symmetric parts of the velocity gradient tensor. According to this definition, a vortex is defined when the eigenvalue, $\lambda_2$, is less than zero. Figure 20 depicts the vortex shed by the blades, on a vertical plane, for the $\lambda$ of 6.67 and 4.17. For the $\lambda$ of 4.17, the tangle of vortices shed by the nacelle and the blade is captured better by LES than uRANS. From Figure 20, it can be seen that the vortex diffusion is reduced by applying large eddy

simulation. Moreover, small scaler turbulent structures are also better resolved by the LES approach.

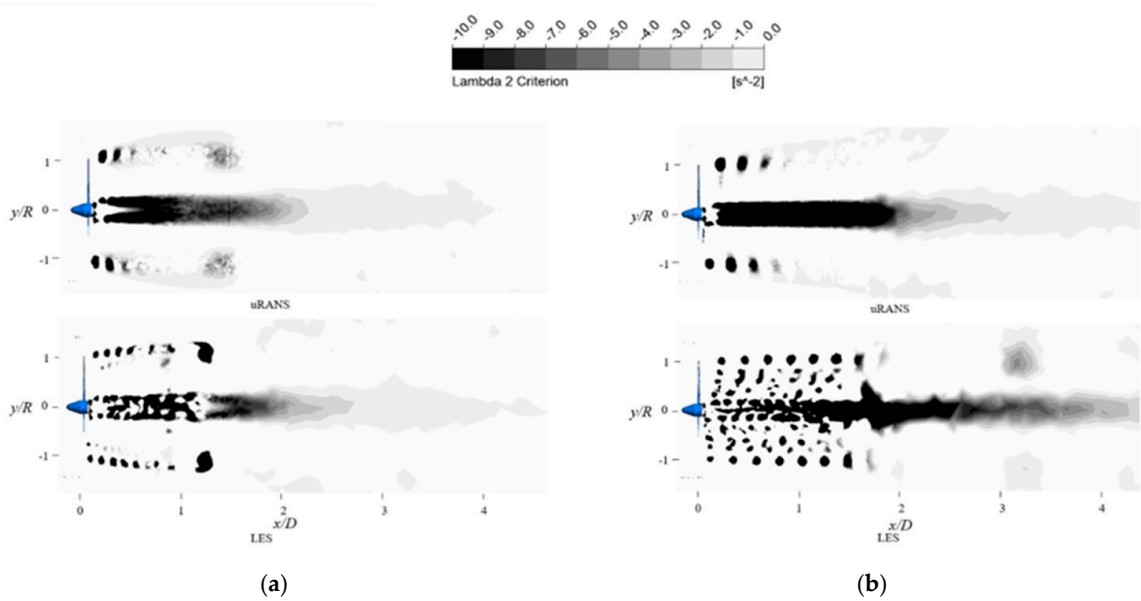

(**a**)  (**b**)

**Figure 20.** $\lambda_2$ criterion plot: (**a**) TSR = 6.67; (**b**) TSR = 4.17.

*3.5. Comparison of the Computational Resources Required by uRANS and LES*

The computational cost associated with the uRANS and LES modeling techniques is discussed in this section. The computational cost comparison is presented here for the $\lambda$ = 6.67. Since the same higher time-step converged solutions were used for both uRANS and LES, the computational cost is compared when the blade completes four revolutions at low time-step ($3.92 \times 10^{-4}$ s) with both uRANS and LES techniques. Simulations were performed in a high-processing computing facility with 32 cores per host. The bar plot in Figure 21 compares the wall time (h) obtained using both the modeling techniques. Here, wall time means the number of hours it takes to complete four revolutions of the wind turbine blade. As expected, LES takes longer than uRANS as it requires higher computational resources. It can be commented that by expending computational cost, which is not drastically higher than the uRANS, better flow characteristics of near wake can be obtained using the LES approach.

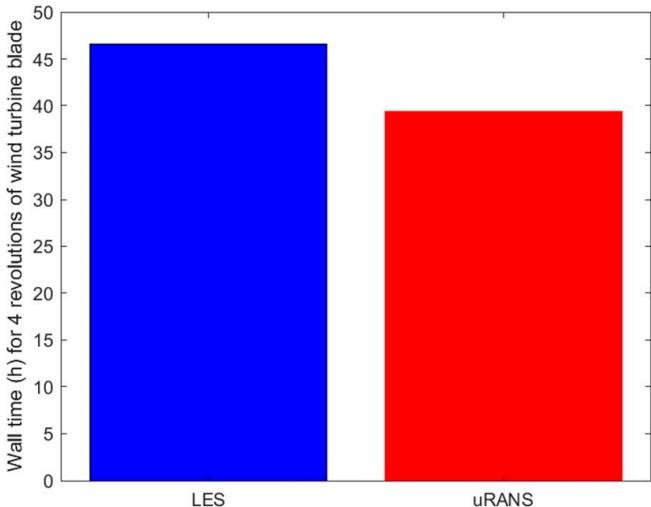

**Figure 21.** Wall time comparison of uRANS and LES for the wind speed case of 15 ms$^{-1}$ ($\lambda$ = 6.67).

## 4. Conclusions

A critical investigation is carried out to study the aerodynamic performance of the LES and uRANS flow modeling approaches. The Phase-III Mexnext (New) MEXICO rotor is chosen to carry out this numerical study, as it gives exhaustive information about the rotor measurements and flow field. The numerical results obtained using the two modeling methods are compared with the experimental results of pressure distributions, along with the flow variables downstream of the blade for axial, radial, and azimuthal traverses.

From this study, the following conclusions can be drawn:

1.  The inception of flow separation for the stall condition of the blade is predicted accurately by the LES approach than by the uRANS approach;
2.  When $\lambda = 10$, the azimuthal traverse of the radial and tangential components of velocity in the inboard region are not predicted more accurately by both of the modeling approaches;
3.  The fluctuations in the wake when the blade operates at $\lambda = 4.17$ are properly captured by the LES approach;
4.  The wake gets recovered faster in the uRANS approach than in the LES method;
5.  LES can capture the small-scale vortices shed by the blade much more accurately.

The results presented in this research have given important insights into the near and far wake characteristics that will be useful in wind farm optimization. In our future works, presented results will be valuable in modeling wind turbines using actuator line modeling. Other important conditions such as different atmospheric boundary layer conditions and turbulent intensities will also be investigated in the future using machine learning techniques.

**Author Contributions:** S.P.: conceptualization, data curation, formal analysis, visualization, investigation, validation, software, writing—original draft. I.F.S.A.K.: data curation, conceptualization. E.Y.K.N.: supervision, writing—review & editing. All authors have read and agreed to the published version of the manuscript.

**Funding:** This research received no external funding.

**Institutional Review Board Statement:** Not applicable.

**Informed Consent Statement:** Not applicable.

**Data Availability Statement:** Not applicable.

**Acknowledgments:** The data used have been supplied by the consortium which carried out the EU FP5 project Mexico: 'Model rotor EXperiments In COntrolled conditions'. The consortium received additional support to perform the New Mexico measurements from the EU projects ESWIRP and INNWIND.EU.

**Conflicts of Interest:** The authors declare no conflict of interest.

## Nomenclature

| | |
|---|---|
| $C_p$ | Pressure coefficient |
| $D$ | Rotor diameter (m) |
| $f_i'$ | Body force per unit mass (N/kg) |
| $k$ | Turbulence kinetic energy (m$^2$/s$^2$) |
| $L_s$ | Mixing length of sub-grid scales (m) |
| $p$ | Pressure (Pa) |
| $P$ | Pressure on the blade (Pa) |
| $P_\infty$ | Atmospheric pressure (Pa) |
| $\bar{p}$ | Mean component of pressure term (Pa) |
| $u$ | Velocity (m/s) |
| $\overline{u_i}$ | Mean velocity along x-axis (m/s) |
| $\overline{u_j}$ | Mean velocity along y-axis (m/s) |
| $u_{ref}$ | Free-stream air velocity (m/s) |

$U_{ax}$    Axial component of velocity (ms$^{-1}$)
$U_{rad}$    Radial component of velocity (ms$^{-1}$)
$U_{tan}$    Tangential component of velocity (ms$^{-1}$)
$V$    Volume (m$^3$)
$x_i$    Distance from the origin along the positive x-axis (m)
$y$    Non-dimensional distance normal to the wall
$\delta_{ij}$    Kronecker delta
$\tau_{ij}$    Stress tensor (N/m$^2$)
$\omega$    Specific dissipation rate (1/s)
$\varepsilon$    Turbulent dissipation rate (m$^2$/s$^3$)
$\mu$    Dynamic viscosity (Ns/m$^2$)
$\Psi$    Azimuthal angle of the blade ($^\circ$)
$\rho$    Density (kg/m$^3$)
$\zeta_n$    Non-dimensional vorticity
$\zeta$    Local vorticity (s$^{-1}$)
$\lambda$    Tip-speed ratio

**Abbreviations**

| | |
|---|---|
| CFD | Computational Fluid Dynamics |
| uRANS | unsteady Reynolds-Averaged Navier Stokes |
| LES | Large Eddy Simulation |
| MEXICO | Model Experiments In Controlled Conditions |
| NREL | National Renewable Energy Laboratory |
| SST | Shear Stress Transport |
| DES | Detached-Eddy Simulation |
| DDES | Delayed Detached-Eddy Simulation |
| RST | Reynolds Stress Transport |
| FEM | Finite Element Method |
| ALE-VMS | Arbitrary Lagrangian-Eulerian Variational Multiscale |
| HAWTs | Horizontal Axis Wind Turbines |
| WALE | Wall-Adaptive Local Eddy Viscosity |
| SIMPLE | Semi-Implicit Method for Pressure Linked Equations |
| MRF | Multiple Rotating Frame |
| SGS | Sub-Grid Scale |
| DNS | Direct Numerical Simulation |
| MAE | Mean Absolute Error |

**Subscripts**

| | |
|---|---|
| $i, j, k$ | 1,2,3 or $x, y, z$-directions |
| ax | Axial |
| rad | radial |
| tan | tangential |

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
