# Peer review of "On the Accuracy of uRANS and LES-Based CFD Modeling Approaches for Rotor and Wake Aerodynamics of the (New) MEXICO Wind Turbine Rotor Phase-III"

_energies, doi:10.3390/en14165198_

Round 1

Reviewer 1 Report

The developed idea could be interesting, but many parts require to be examined and rearranged.

The description of the methodology is not very clear and it is very hard to read. 

I suggest the authors reorganize the paper and resubmit it again. 

Reviewer 2 Report

This paper presents a numerical study on the aerodynamic performance of a wind turbine for which exhaustive experimental results are available. The numerical results obtained using two modeling methods, uRans and LES, are compared with the experimental results. 
These are results providing important information on the characteristics of the near and far wake, which I believe will be useful in the optimization of wind farms.
The paper is well structured and well written. I recommend its publication as is in the ENERGIES journal. 

Reviewer 3 Report

Dear Authors

I find your paper very interesting. A massive work has been done. It is really impressive.

Below I present general inaccuracies found. The correction of them should improve the quality of paper.

General comment. I think the Authors should introduce some description about the measurement technique. This would make the paper more complete for a reader. Now it evidently lacks.

P4 (Section 2.2). General advice. In the future I advise to use formula for the turbulence intensity
I=16*Re^(-0.125) [%] for BC. Then you would know how intensive it is and then it cannot have been chosen a’priori.

General remark. The Authors nothing say about justifications of the mesh used to computations. Please justify your computational/numerical results.

General remark. No section 2.4.2 is found. Must be revised.

Fig.6. The quality of partial figures is very poor. I suggest to lie it vertically (MDPI does not limit number of pagres). The poor quality generally concerns more figures. Please enlarge them for a higher readiness.

Fig.7. Did the Author try approximating the experimental points with the smoothed line? This would be more comparable to numerical results. Consider replacing broken lines with smoothed line.

The description contained in section 3.1 should be moved up before Fig.7. Now it looks strange.

Fig.18. Why does r/R exceed value 1? Explain in the text.

Fig.19. Enlarge colour maps. Now they looks unreadable. Additionally, why are they (vorticity) with unit [-]?

Fig.20. Enlarge colour maps. Now they looks unreadable.s

P21. A ‘wall time’ term should be explained in the text for a reader’s convenience. Additionally, I strongly suggest to present absolute values of hours required to compute 4 revolutions.

Round 2

Reviewer 1 Report

The revision is appropriate